# Augmenting the Efficacy of the Initial Patient Visit to the Stroke Prevention Clinic: A Quality Improvement Project

**DOI:** 10.3390/jcm14248780

**Published:** 2025-12-11

**Authors:** Anastasia Howe, Sunpreet Kaur Cheema, Farah Saleh, Thomas Jeerakathil, Pamela Mathura, Mahesh Kate

**Affiliations:** 1Division of General Internal Medicine, Department of Medicine, University of Alberta, Edmonton, AB T0B4J2, Canada; amhowe@ualberta.ca; 2Faculty of Medicine and Dentistry, University of Alberta, Edmonton, AB T6G 2R7, Canada; sunpree1@ualberta.ca (S.K.C.);; 3Division of Neurology, Department of Medicine, University of Alberta, Edmonton, AB T6G 2B7, Canada; 4Department of Medicine, University of Alberta, Edmonton, AB T6G 2B7, Canada

**Keywords:** stroke prevention, transient ischemic attack, quality improvement, referral triage, pre-visit intervention, secondary stroke prevention

## Abstract

**Background:** Referrals to the Stroke prevention clinic with incomplete preliminary investigations decrease clinic capacity due to additional workload and the need for multiple follow-ups. We aimed to improve the efficacy of the initial visit by increasing the completion rate of vascular imaging in a quality improvement (QI) project. **Methods:** This is a quasi-experimental study with three phases: Phase 1: Surveillance; Phase 2: Stakeholder feedback-informed intervention development (physicians and clinic staff); and Phase 3: Intervention. Interventions included a new standardized specific physician triage form listing required investigations (brain imaging, vascular imaging, cardiac tracing), and a nurse-led pre-visit via telephone. The primary outcome measure was the completion of vascular imaging by the time of visit, assessed using multivariable logistic regression adjusted for age (in years), sex, and triage category. **Results**: The study’s inclusion criteria were met by 397 patients, with a mean age of 67.7 ± 13.2 years; 47.8% were female, and 62.9% (250) were diagnosed with vascular events. An increase in vascular imaging before the initial visit was observed in Phase 3 (148/199, 75.5%) compared to Phase 1 (121/198, 61.1%), with an adjusted Odds ratio of 1.77 (95% CI 1.2–2.8; *p* = 0.01). A trend toward fewer follow-up visits was observed in Phase 3 (23.1%) compared with Phase 1 (31.8%; *p* = 0.052). **Conclusions**: Implementing a standardized triage process and a nurse-led pre-visit may improve completion of vascular imaging before patients visit the stroke prevention clinic. Further QI studies are required to improve the completion rate of rhythm monitoring in this patient group to enable early detection and management of atrial fibrillation.

## 1. Introduction

A transient ischemic attack (TIA) and minor ischemic stroke are associated with a risk of recurrent stroke. This risk is high in the first 30 days, with a frontloaded burden in the first 48 h [1,2]. The days following a TIA or minor stroke represent a critical window for secondary stroke prevention. The EXPRESS study demonstrated that urgent assessment and treatment of TIA and minor stroke in outpatient settings reduced the 90-day risk of recurrent stroke by approximately 80% [3]. The benefit was attributed to the prompt initiation of antiplatelet therapy, rapid vascular imaging, and vascular risk management. To meet this need, health systems worldwide have developed dedicated Stroke Prevention clinics (SPC) [4,5,6,7]. SPCs have become a practical alternative to routine hospitalization, offering expedited evaluation while preserving inpatient resources [8,9].

In many regions, increased referrals to the clinics, limited access to the neurologists, and imaging have translated into longer wait times for assessment, despite evidence that delays increase the risk of stroke [10]. Yet, the first clinic visit often falls short of its potential. Patients are sometimes referred to SPC without completion of essential diagnostic tests, such as brain imaging, vascular studies, or cardiac rhythm assessment with electrocardiogram (ECG) or Holter monitoring. In such cases, clinicians are unable to make immediate treatment decisions, leading to follow-up visits or prolonged diagnostic timelines [10,11,12,13].

This incomplete first patient visit reduces efficiency and may delay the start of preventive therapy at a time when the risk of recurrent stroke is the highest. One strategy to overcome this challenge is to embed a structured pre-clinic planning visit (pre-visit) process. In the setting of chronic disease management, such as diabetes, pre-visit may serve as a better triage tool, improve patient engagement and compliance [14,15,16]. The concept of pre-visit planning has evolved, initially focusing on patient triage and now including patient assessment and management. If pre-visit is applied to SPC, this approach could ensure patients arrive at their appointment with the investigation completed, enabling immediate initiation of evidence-based secondary prevention. Aligning such interventions with Canadian stroke best practice recommendations may help close the gap between guideline expectations and real-world practice [14,15,16].

This quality improvement (QI) study aimed to evaluate the impact of a standardized triage process and nurse-led pre-visit intervention on investigation completion rates, secondary prevention timelines, and follow-up visit frequency in patients referred to an SPC. We hypothesized that in patients referred to the SPC, an intervention comprising a standardized triage process and a structured nurse-led pre-visit would be associated with a higher rate of completion of vascular imaging by the time of the first patient visit compared to the workflow without the study intervention.

## 2. Materials and Methods

This study utilized a pre-post experimental design to evaluate the impact of a multi-component QI intervention on the initial outpatient assessment in the SPC. The QI study followed the plan-do-study-act (PDSA) cycle. Ethics approval for this quality improvement initiative was obtained through the Health Research Ethics Board—Health Panel (Pro00136414) as a research study for retrospective chart review. The project adhered to the ethical principles governing quality improvement research.

### 2.1. Study Setting and Population

The SPC at the University of Alberta Hospital (Edmonton, Alberta, Canada) is a high-volume tertiary care outpatient clinic that receives approximately 1300 new referrals annually. Referral sources include emergency departments (EDs), primary care providers (PCPs), and other subspecialty clinics. The reasons for referrals could include rapid assessment, diagnosis of cerebrovascular events (such as TIA, arterial and venous strokes), management advice, and evaluation of silent/asymptomatic lesions (such as infarcts or hemorrhages). The SPC physician triages the referral based on the patient’s symptoms and investigations completed in accordance with the Edmonton Zone Transient Ischemic Attack and Non-disabling Stroke Management Algorithm [17]. Based on the triage category, patient appointments are booked (Figure 1):

Inclusion Criteria: Patients were eligible for this study if they were aged 18 years or older and had been referred for initial evaluation (new referral) at the SPC between January and September 2024. 

Exclusion criteria: Patients who were follow-ups after a hospital admission for stroke from an established stroke unit (these were identified by the triage physician as already completed investigations as per standard of care and were assigned triage category as routine to be seen within 3 months), or who had previously been evaluated (in person or via Tele-stroke) by a stroke neurologist, were excluded. By focusing on first-time outpatient evaluations, the study targeted the population most vulnerable to delayed diagnosis and missed opportunities for secondary prevention.

A structured chart review was performed of patients who attended the clinic at the end of the study period in both the surveillance and intervention phases. This formed the screening population, and, based on the inclusion and exclusion criteria described above, patients were included. The data abstraction was performed by independent reviewers (AH, SC, FS).

### 2.2. Operational Definitions

To ensure consistent data capture and interpretation, several operational definitions were established a priori. Essential investigations were defined as brain imaging (either CT or MRI of the brain), vascular imaging (CT angiography or Carotid Doppler ultrasound), and ECG (performed either in the ED or in an outpatient setting prior to the clinic visit within 3 months of the event). The term “cerebrovascular event” encompasses TIAs and minor strokes (major ischemic strokes were usually admitted to the hospital, therefore were not part of this cohort unless they were admitted previously to a rural center and a stroke neurologist was not involved in care previously).

“Missed prevention opportunity events” were defined as either a recurrent stroke or TIA occurring during the period a patient waited to be seen in the clinic; investigations ordered and completed by the referring provider that were not acted upon for over a week, and that ultimately changed management; or delays exceeding 30 days in initiating secondary prevention therapy (in the form of antiplatelet or anticoagulant). The clinical outcome measures were analyzed as a composite due to the lower sample size and the lower likelihood of the event rates. “Initiation of secondary prevention” was defined as the start of antiplatelet or anticoagulant therapy within 48 h of assessment.

### 2.3. Study Design and Phases

The study followed a quasi-experimental pre-post experimental design with three sequential phases, each building on insights gained from the preceding phase. This approach enabled iterative refinement of the intervention before launch and allowed for careful monitoring of both clinical and process outcomes.

Phase 1: Surveillance (January–March 2024)

During the three-month pre-intervention period, all new outpatient referrals were screened. A total of 198 patients (61.9% of screened referrals) met the study criteria and were included. The definition of new referrals was one in which a neurologist or a stroke physician was not consulted in the ED or during the hospital stay. For each patient, the research team prospectively abstracted data from the electronic medical record (EMR), including completion of essential diagnostic investigations, the time from referral to investigation completion, and the time to initiation of secondary prevention therapies, such as antiplatelet or anticoagulant medications. Missed prevention opportunity events occurring before the first clinic visit were also recorded. Demographic data, including age, sex, and referral source, were captured, along with the final diagnosis of either a cerebrovascular event or a stroke mimic (as defined by the Stroke specialist during the clinic appointment). The baseline data collection phase provided a detailed understanding of the existing clinic workflow, the completeness and timing of diagnostic testing, and gaps in early secondary prevention, forming the foundation for the subsequent intervention.

Phase 2: Stakeholder-Informed Intervention Development (April–May 2024)

Following baseline data collection, the research team engaged with SPC stakeholders, including stroke neurologists, nursing staff, clinic administrative staff, and diagnostic imaging personnel. A multi-disciplinary meeting was organized to review baseline findings and to co-design the intervention. Drawing on principles from quality improvement literature, the team emphasized pre-appointment coordination and system-level decision support to ensure timely, complete, and safe care. This phase allowed input from front-line clinicians who were familiar with the operational challenges. It helped tailor the intervention to the clinic’s real-world constraints, including patient volume, staffing patterns, and diagnostic availability.

Phase 3: Intervention Implementation (July–September 2024)

The intervention combined several interlinked components designed to improve the efficiency and safety of the first SPC visit. Standardized physician triage forms were implemented to assess if core investigations were completed. If tests were not completed, we assessed the need for a pre-visit appointment.

*Nurse-Led Pre-Visit Phone Call*: Patients identified as having incomplete workups were contacted by an SPC nurse at the earliest available time. During the phone call, the nurse reviewed the patient’s EMR to verify prior investigations, clarified clinical history, and initiated outstanding tests recommended by the triage Stroke specialist, such as Carotid Doppler ultrasounds, brain imaging, bloodwork (for diabetes screening or lipid panel assessment) or Holter monitoring referral. Where clinically appropriate, patients were reminded to start or resume secondary prevention therapies in collaboration with the supervising neurologist (MK). All interventions and orders were documented in the EMR to ensure traceability and continuity of care. The pre-visit process required 15–20 min per patient. Following implementation, data was collected from 199 new referrals who met the same eligibility criteria as the baseline cohort. The same variables were captured, including diagnostic completeness, time to investigation, initiation of secondary prevention, need for follow-up visits, and missed prevention opportunity rates. The consistency of data collection before and after intervention allowed a robust comparison of outcomes and process metrics (Figure 2).

A post-implementation review was conducted with the same multidisciplinary stakeholders to evaluate the intervention’s impact and sustainability. Staff satisfaction was qualitatively assessed.

### 2.4. Outcome Measures

The primary outcome of the study was the proportion of patients who had completed vascular imaging (CT angiography for the head and neck or carotid Doppler) before their first SPC visit, as this was the investigation over which the SPC team had the most control to book a timely appointment. Secondary outcomes included the proportion of patients who completed all three core investigations, the mean time to completion of investigations, the number of follow-up visits required, the time from referral to initiation of secondary prevention, and the rate of missed prevention opportunities.

### 2.5. Process Measures

In addition to clinical outcomes, the study tracked process and balancing measures to assess operational impact. These included the total number of nurse pre-visit calls completed, the types of investigations initiated or coordinated during these calls, and the percentage of patients who resumed or started secondary prevention therapy before their first SPC visit. Clinic resource use and staff workload were monitored qualitatively, allowing for an evaluation of whether the intervention created an additional burden or improved workflow efficiency.

### 2.6. Sample Size Calculation and Statistical Analysis

We aimed to increase the proportion of patients completing vascular imaging prior to the clinic visit. Based on a previous regional study, approximately 61.4% receive urgent carotid imaging. To improve the rate of vascular imaging by an absolute 14%, we will need a sample size of 171 per group, with 80% power and 5% alpha, for a total of 342 patients. To accommodate no-show visits, we aimed for a sample size of 190 referrals per group [18].

The two comparator groups were the pre-intervention and intervention groups. Categorical variables, including sex, diagnosis of cerebrovascular event, referral source, triage category, essential investigation done and follow-ups required, were expressed as frequencies and percentages. Continuous variables age (years), symptom onset to stroke clinic appointment (days) and symptom onset to completion of investigations (days) were expressed as mean ± SD if the variable was distributed normally and median (IQR) if the variable was distributed non-normally. Categorical variables were compared between the pre-intervention and intervention groups using the Chi-Square test for univariable analysis and logistic regression for multivariable analysis. The mean age (in years) was compared between the pre-intervention and intervention groups using an unpaired t-test. Symptom onset to stroke clinic appointment (days) and symptom onset to completion of investigations (days) were expressed as medians (interquartile ranges, IQR) and compared using the Mann–Whitney U test. To assess the effect size of the intervention on completion of vascular imaging, a multivariable logistic regression was performed to obtain the odds ratio. The odds ratio was adjusted for a fixed set of variables, which were chosen a priori, including age (in years), sex (%), and triage category. Significance was determined with a threshold of *p* < 0.05. All statistical analyses were performed using Stata 18.0 BE (STATAcorp LLC, College Station, TX, USA).

## 3. Results

### 3.1. Patient Cohorts, Demographics

Over the six-month study period, 397 new outpatient referrals to the Stroke Prevention Clinic (SPC) met the inclusion criteria and were analyzed. During the pre-intervention period, 320 referrals were screened, of which 198 patients (61.9%) met the inclusion criteria. During the intervention period, 453 referrals were screened, and 199 patients (43.9%) were included. The demographic and referral characteristics were similar across groups (Table 1), except that the median days to appointment were prolonged by almost 10 days during the project intervention phase.

### 3.2. Diagnostic Workup

Implementation of the pre-visit triage process and nurse-led intervention was associated with an improvement in diagnostic test completion (Table 2). The completion rate of vascular imaging increased from 61.1% in the pre-intervention phase to 75.5% in the intervention phase (OR 1.84, 95% CI 1.2–2.8; *p* = 0.005). This improvement persisted in multivariable analysis (OR 1.77, 95% CI 1.2–2.8, *p* = 0.01), adjusted for age, sex, and triage category. The completion of vascular imaging contributed to a corresponding rise in the proportion of patients who completed all three core diagnostic investigations—brain imaging, vascular imaging, and electrocardiography (ECG)—from 55% to 65% (*p* = 0.049). Completion of brain imaging remained consistently high in both phases (>90%), with no statistically significant difference (*p* = 0.7). In contrast, Holter monitoring demonstrated persistently low completion rates and declined further following implementation (35.8% vs. 25.1%; *p* = 0.02), suggesting ongoing barriers to timely cardiac rhythm assessment despite overall gains in diagnostic efficiency. There was a trend toward a lower median number of days required to complete the diagnostic tests in the post-intervention phase than in the pre-intervention phase in linear regression analysis (beta = −6.1, 95% CI = −12.6–0.4, *p* = 0.06).

#### Nurse-Led Pre-Visit Subgroup

During the intervention phase, a total of 69 (34.7%, n = 199) patients were triaged for pre-visit review, of whom 54 (78.3%, n = 69) successfully completed the structured nurse-led pre-visit call. Fifteen (21.7%) patients could not be reached for the pre-visit. The pre-visit was completed at a median of 52 (5–66) days after symptom onset. Among these patients, 76.8% (53/69) completed all core diagnostic investigations prior to attending the Stroke Prevention Clinic (SPC). Brain imaging was achieved in 97% (67/69) of cases, reflecting excellent adherence to imaging recommendations. Carotid Doppler studies were the most frequently ordered investigation 89% (n = 61/69). All telephone calls required <25 min per call.

Notably, three (5.5%, n = 54) pre-visit cases revealed significant carotid stenosis (1: above 70% symptomatic; 2: above 70% symptomatic; 3: 50–70% asymptomatic) requiring urgent vascular intervention; two patients underwent successful revascularization within 7–8 days of the pre-visit, while one case- classified as a possible migraine with incidental asymptomatic stenosis- was managed conservatively. These cases highlight the benefits of early triage, as similar findings would likely have been delayed by approximately two months under the standard referral and appointment pathway.

Holter monitoring requests were infrequent (2 patients), reflecting persistent system-level barriers such as limited access to community testing and prolonged wait times at tertiary centers. Four patients required no additional investigations after pre-visit review, as all necessary tests had already been completed. Fourteen (25.9%) patients were advised to initiate or resume secondary prevention prior to their scheduled SPC appointment, demonstrating the proactive clinical and preventive impact of the nurse-led coordination model. Nineteen (35%) patients were subsequently on clinic appointment visit diagnosed as a stroke mimic. No adverse events were reported after nurse-led pre-visit clinic-initiated treatment.

### 3.3. Timeliness of Care

Delays exceeding 48 h in initiating secondary prevention therapy decreased from 27 cases (13.6%) to 12 cases (6%), suggesting a trend toward timelier management. However, it is unclear whether the QI project intervention alone was responsible for that effect, as there are confounding variables including referral source, symptom severity and comorbidities. These variables were not systematically abstracted in the chart review.

### 3.4. Clinic Workload Efficiency

The intervention was associated with a reduction in the need for repeat clinic visits due to incomplete diagnostic workup. In the pre-intervention phase, 31.8% of patients required at least one follow-up visit, compared to 23.1% in the post-intervention phase (*p* = 0.052), indicating a trend toward improved first-visit efficiency. Among participants in the structured nurse-led pre-visit process, only 15% required additional appointments, underscoring the effectiveness of triage and coordination in optimizing clinic efficiency.

SPC Staff (n = 15) feedback demonstrated unanimous agreement that the pre-visit model enhanced overall patient care, with a mean satisfaction score of 4.3 out of 5. One respondent reported a relative increase in clinic workload. However, there was no increase in administrative burden. We did not capture the time required for scheduling the nurse-led pre-visit, scheduling investigations, and the time taken for the Doppler test. The intervention was implemented entirely through the reorganization of existing staff roles and resources, without requiring additional funding, highlighting its sustainability and practicality within routine clinical operations.

### 3.5. Patient Safety

Patient safety indicators demonstrated overall improvement following implementation of the nurse-led pre-visit intervention. Missed prevention opportunity events—defined as cases of delayed diagnosis or delayed initiation of secondary prevention—declined from 7.5% in the pre-intervention period to 4.3% post-intervention. However, this difference did not reach statistical significance (*p* = 0.184).

## 4. Discussion

This quasi-experimental study demonstrated that implementing a structured triage process and a nurse-led pre-visit telephone intervention may improve the completion rate of vascular imaging. The vascular imaging was completed in a timely manner. The study addresses the long-standing question of the effectiveness of the first visit, in which patients often arrive without the essential investigations required for definitive risk stratification and initiation of secondary prevention. In a systematic review, pre-visit planning was considered effective in clinical practice [19].

*Diagnostic Preparedness Enables Definitive Visits*: A major consequence of the incomplete workups is the inability to make definitive treatment decisions at the first visit. This creates a cascade of inefficiencies, such as follow-up appointments, additional test bookings, and care fragmentation. Before the intervention, nearly half of the patients lacked one or more core tests, such as brain imaging, vascular imaging, or even a basic ECG, at the initial visit. This challenge has been consistently reported in Canadian and international prevention clinic models, where incomplete work-ups delay diagnosis and treatment initiation [4,20]. With the triage protocol and nurse-led coordination, our intervention resulted in a 10% absolute increase in complete diagnostic workups. By improving the proportion of patients arriving with completed investigations from 55% to 65% and from 61% to 75% for vascular imaging alone, the intervention may have enhanced clinicians’ ability to diagnose and treat patients in a single visit. However, cardiac rhythm monitoring with Holter Monitoring was lower and even decreased during the intervention phase. This suggests the need for a further QI project addressing core investigations. It will be important to involve community stakeholders, including increasing access to patch monitors. This would include advocacy efforts at the regional and provincial levels, given variable access to patch monitors.

*Nurse-led Pre-visit:* Nurse-led pre-appointment visits have been successfully implemented in perioperative settings, urogynecology clinics, chronic disease management, and patient education [21]. In a systematic review, pre-visit reduced cancellations of surgeries in orthopedic practices [22]. In another systematic review, nurse-led pre-visit planning has been shown to enhance the quality of patient care [19]. This finding aligns with the findings of Harrington et al., who concluded that pre-appointment management should be a key strategy for reducing health care costs, addressing personnel shortages, and improving access [23]. Similar results from SOS-TIA and other rapid-access clinic models also support our findings, as they too showed a reduction in recurrent stroke risk with structured care pathways [3,4].

Nurse-led telecare is an acceptable option for patients during the post-acute phase and may also improve patient-related outcomes measures [24]. In another study, advanced practice registered Nurse-led clinics for post-stroke patients were associated with reduced unplanned 30-day re-admissions [25]. Our study aligns with the above results and demonstrates increased completion rates of vascular imaging.

*Enhancing Safety Through Early Detection*: The most striking finding from the SPC pre-visit encounter was that 13.3% of patients had clinically significant findings which were identified before their clinic appointment. These included newly detected severe carotid stenosis, critical lipid abnormalities, or cases of stroke mimic misdiagnosis. In 3 cases, the vascular surgery referral was expedited and completed within 2 weeks of the referral. A multicenter randomized controlled trial established that successful CEA for asymptomatic patients younger than 75 years of age reduces 10-year stroke risks. Half of this reduction is in disabling or fatal strokes [26]. For multiple patients who were not started on antiplatelet therapy before the first SPC visit or were hesitant to initiate therapy, the antiplatelet therapy was initiated days before the first SPC visit through a nurse-led telephone appointment. They represent an efficient approach to secondary stroke prevention through proactive care. This approach mirrors similar strategies in public health and primary care. For example, structured pre-visit planning for complex diabetes patients improved prioritization and reduced missed care opportunities [27].

*Strengths and Limitations*: A strength of the study was its demonstration that scalable low-cost process changes can improve both patient safety and clinic workflow without overwhelming diagnostic services. Carotid ultrasound demand increased; however, the clinic’s US technicians accommodated the volume without delays, and staff reported greater satisfaction with the workflow.

This study was conducted in a single academic stroke center, which may limit its generalizability to community or rural care settings. Patients admitted to stroke units were excluded; this may have introduced selection bias and increased the proportion of incomplete core investigations. Additionally, while the pre-post design offers valuable insights, it is susceptible to temporal confounding. Underutilization of cardiac monitoring remains a challenge. Despite gains in vascular imaging, completion rates for Holter monitoring remained low. This reflects the logistical challenges common in the stroke clinics. Future efforts should aim to integrate more effective ambulatory cardiac monitoring options directly into the SPC workflow.

While patient demographics were stable across pre- and intervention phases, the results could have been influenced by concurrent system-level changes. A modified electronic referral form was introduced during the study period, which included optional prompts for referring providers to indicate whether core investigations, such as brain imaging, vascular imaging, ECG, and basic laboratory tests, had been completed. This change aimed at standardizing the information available at triage and could have contributed to the observed improvements in investigation completion rate. Similar initiatives in other settings have shown that structured electronic tools and decision support can improve information quality and guide providers to make more complete referrals, which in turn supports more timely care [28]. The median number of days to the appointment was delayed during the intervention phase of the QI project compared to the surveillance phase. While no patient-related or systemic factors could be identified, the delay was not related to the project intervention. The Nurse-led Pre-visit was organized after the patient’s clinic appointment was planned.

Additionally, while delays in secondary prevention improved, a small proportion of patients still faced waits of more than 48 h, underscoring the importance of initiating therapy directly in Emergency department or primary care prior to SPC referral.

Finally, the study did not detect changes in long-term clinical outcomes, such as recurrent stroke or mortality.

## 5. Conclusions

Krueger et al. projected an increase in the number of individuals experiencing stroke by 2038 in Canada, highlighting the urgency of optimizing secondary prevention strategies [29]. A systematic review of global stroke guidelines strongly recommends establishing an etiological diagnosis and managing key risk factors [16]. This quality improvement initiative demonstrates that a structured triage algorithm combined with a well-established pre-visit process may enhance the quality, safety, and efficiency of outpatient stroke care. By shifting diagnostic review and coordination upstream in the care pathway, the intervention increased completion rates of essential investigations, such as vascular imaging, and reduced follow-up visits. Since our study is a QI project and non-randomized, the results will be considered exploratory.

This intervention was low-cost, scalable, and well-integrated into the existing clinical workflows. The model empowered the nursing staff to play a proactive role in patient care, aligning with non-financial incentives as outlined in systematic reviews [30].

Although barriers remain, particularly regarding access to Holter monitoring and adoption of EMR referrals, the early impact of this intervention is promising. With 13% of pre-visit patients identified as having urgent findings requiring management before their scheduled clinic encounter, this approach is potentially “stroke-preventing”.

## 6. Future Directions

Future efforts could focus on automating the triage tools within electronic health records, integrating extended cardiac monitoring options, and exploring multicenter implementation to validate generalizability.

## Figures and Tables

**Figure 1 jcm-14-08780-f001:**
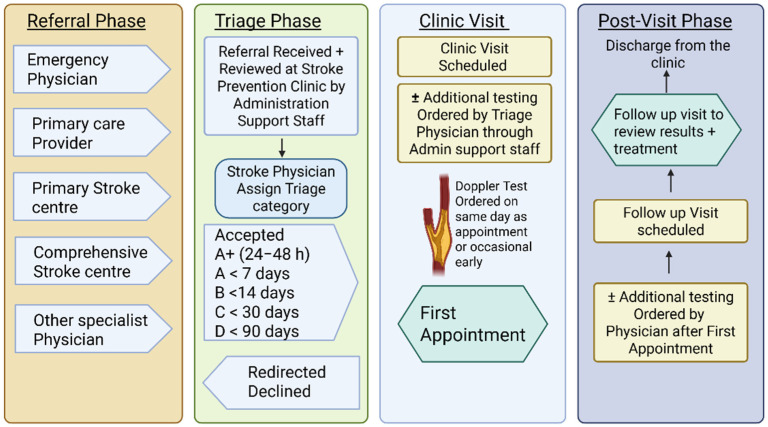
Stroke Prevention Clinic (SPC) pre-intervention workflow, triage, and patient visit. Created in BioRender. Kate, M. (2025) https://BioRender.com/buv1zgx (accessed on 8 December 2025).

**Figure 2 jcm-14-08780-f002:**
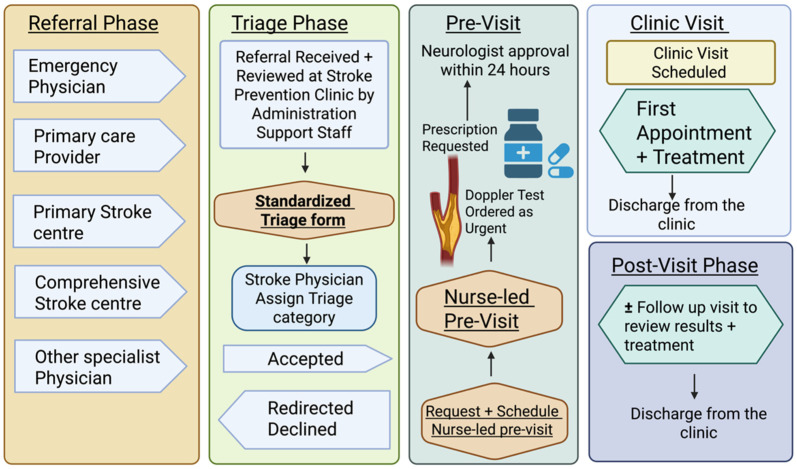
Stroke Prevention Clinic (SPC) workflow, triage, and patient visit during the intervention phase. Created in BioRender. Kate, M. (2025) https://BioRender.com/94l76nj (accessed on 5 December 2025).

**Table 1 jcm-14-08780-t001:** Patient Characteristics Pre- vs. post-intervention.

Characteristic	Pre-Intervention (n = 198)	Intervention (n = 199)	*p* Value
Mean age, years (±SD)	67 ± 13.5	68.4 ± 13.5	0.29
Female, n (%)	99 (50.0%)	91 (45.7%)	0.45
True Cerebrovascular event, n (%)	128 (64.6%)	122 (61.3%)	0.55
Stroke Mimics, n (%)	70 (35.4%)	77 (38.7%)	
Referral sources: ED, n (%)	124 (62.6%)	133 (66.8%)	0.07
Referral Source: PCP, n (%)	53 (26.7%)	41 (20.6%)	
**Triage category**			
A+ Emergent, n (%)	6 (3.0%)	6 (3.0%)	0.15
A Urgent, n (%)	27 (13.6%)	14 (7.0%)	
B Semi-urgent, n (%)	63 (31.8%)	61 (30.7%)	
C Non-urgent, n (%)	65 (32.4%)	66 (33.2%)	
D Routine, n (%)	37 (18.7%)	52 (26.1%)	
**Median Days to appointments according to the triage category after Triage**			
A+ Emergent	5 (1–8)	6.5 (4–13)	
A Urgent	7(5–20)	11 (5–14)	0.04
B Semi-urgent	39 (30–50)	43 (26–61)	
C Non-urgent	49 (42–53)	57 (42–64)	
D Routine	94 (50–124)	72.5 (61–93.5)	
**Median (IQR) Days to Appointment after Triage**	46 (29–56)	56 (28–69)	0.008
Follow-up Visits, n (%)	63 (31.80%)	46 (23.10%)	

PCP—Primary Care Provider, ED—Emergency Department.

**Table 2 jcm-14-08780-t002:** Diagnostic Test performed prior to first visit in the Pre- vs. Intervention groups.

Investigation	Pre-Intervention (n = 198)	Intervention (n = 199)	*p* Value
Vascular Imaging n (%)	121 (61.1%)	148 (75.5%)	0.005
Brain Imaging (MRI/CT), n (%)	180 (90.9%)	184 (92.46%)	0.6
Electrocardiography, n (%)	177 (89.4%)	174 (87.43%)	0.5
Holter Monitoring, n (%)	71 (35.8%)	50 (25.12%)	0.02
All Core Investigations, n (%)	109 (55%)	130 (65.3%)	0.049
Median Days to investigation completion	6 (1–34)	2(0–16)	<0.001

## Data Availability

The full data are available upon request.

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
