# Peer review of "Augmenting the Efficacy of the Initial Patient Visit to the Stroke Prevention Clinic: A Quality Improvement Project"

_jcm, 2025, doi:10.3390/jcm14248780_

Round 1
Reviewer 1 Report
Comments and Suggestions for Authors
This is a well-intended quality-improvement study focusing on structured triage and a nurse-led telephone pre-visit to enhance the initial Stroke Prevention Clinic encounter. The topic is clinically relevant, aligns with national stroke guidelines, and fits well within the current movement toward QI-based models for early TIA/minor stroke evaluation However, several methodological inconsistencies, statistical reporting issues, unclear definitions, and areas requiring structural tightening weaken the clarity and scientific rigor of the manuscript. The intervention is promising, but the manuscript needs refinement to ensure interpretability, reproducibility, and scientific validity.

Can be improved by Native English Speaker
Author Response
Reviewer 1
- Major Comments
Title and Abstract
- Title: Accurate but slightly long. Consider clarifying that this is a quality improvement project in the title for indexing clarity.
We have edited the title and added the project to it. Title: Augmenting the Efficacy of the Initial Patient Visit to the Stroke Prevention Clinic: A Quality Improvement Project.
- Abstract:
Methods (Lines 18–24): The design is stated as “pre-post experimental,” but this is a quasi experimental QI design. The distinction should be clearly maintained to avoid over interpretation.
We agree with the reviewer and edited the paragraph to reflect it. The following has been added, “ Methods: This is a quasi-experimental study with three phases: Phase 1: Surveillance; Phase 2: Stakeholder feedback-informed intervention development (physicians and clinic staff); and Phase 3: Intervention.”
Results (Lines 25–33): Provide actual p-values consistently.
The statement “fewer follow-up visits were required… p = 0.052” should be clarified as a trend, not statistically significant.
We agree with the reviewer and edited the line to reflect it. The following has been added, “A trend toward fewer follow-up visits was observed in Phase 3 (23.1%) compared with Phase 1 (31.8%; p = 0.052).”
Conclusions (Lines 31–33): The conclusion is appropriate but could better reflect the modest effect size and the persistent issues with Holter monitoring.
We agree with the reviewer and edited the paragraph to reflect it. The following has been added, Conclusions: Implementing a standardized triage process and a nurse-led pre-visit may improve completion of vascular imaging before patients visit the stroke prevention clinic. Further QI studies are required to improve the completion rate of rhythm monitoring in this patient group to enable early detection and management of atrial fibrillation. “
Introduction
Lines 39–69: The introduction is factually sound but overly long. Consider tightening the historical descriptions and focusing more on the specific local gap that prompted this intervention.
We have broken the introduction line itnto two. We have edited the introduction. The following had edited, “A transient ischemic attack (TIA) and minor ischemic stroke are associated with a risk of recurrent stroke. This risk is high in the first 30 days, with a frontloaded burden in the first 48 hours [1][2]. The days following a TIA or minor stroke represent a critical window for secondary stroke prevention. Studies have consistently shown that early interventions during this high-risk period can reduce the risk of recurrence. The EXPRESS study demonstrated that urgent assessment and treatment of TIA and minor stroke in outpatient settings reduced the 90-day risk of recurrent stroke by approximately 80% [[3]]. The benefit was attributed to the prompt initiation of antiplatelet therapy, rapid vascular imaging, and vascular risk management. To meet this need, health systems worldwide have developed dedicated Stroke Prevention clinics (SPC) designed to provide rapid outpatient assessment, targeted investigations, and an intervention of preventive therapy [4][5][6][7]. SPCs have become a practical alternative to routine hospitalization, offering expedited evaluation while preserving inpatient resources [8,9].”
Lines 47–56: Several statements attribute delays solely to incomplete imaging, but the authors do not provide local baseline evidence until the Results section. Consider previewing your baseline problem magnitude here.
We describe and discuss the magnitude of the problem in a paragraph.
Lines 62–66: The description of pre-visit planning cites chronic disease models but does not sufficiently justify why a similar model is applicable to acute cerebrovascular risk scenarios. A sentence linking the urgency of secondary prevention with pre-visit triage would strengthen rationale.
We have now revised the paragrapgh to develop a link, “ ​​One strategy to overcome this challenge is to embed a structured pre-clinic planning visit (pre-visit) process. In the setting of chronic disease management, such as diabetes, pre-visit may serve as a better triage tool, improve patient engagement and compliance [14][15][16]. The concept of pre-visit planning has evolved, initially focusing on patient triage and now including patient assessment and management. If pre-visit is applied to SPC, this approach could ensure patients arrive at their appointment with the investigation completed, enabling immediate initiation of evidence-based secondary prevention.”
Methods
Design
Line 78: Describe explicitly whether this QI project followed any structure (e.g., PDSA cycles, Model for Improvement). It currently reads as observational rather than iterative.
We have added the following line to reflect that, “ This QI study followed the plan-do-study-act (PDSA) cycle.
Line 80: Ethics approval is mentioned but clarify whether this was classified as QI or research by your IRB.
The following has been added, “ Ethics approval for this quality improvement initiative was obtained through the Health Research Ethics Board – Health Panel (Pro00136414) as a research study for retrospective chart review. The project adhered to the ethical principles governing quality improvement research.”
Setting & Population
Lines 83–104: The inclusion/exclusion criteria need a clearer definition:
We have edited the paragraph for clarity as Inclusion Criteria: Patients were eligible for this study if they were aged 18 years or older and had been referred for initial evaluation (new referral) at the SPC between January and September 2024. Only new referrals were included. Exclusion criteria: Patients who were follow-ups after a hospital admission for stroke from an established stroke unit (these are identified by the triage physician as a standard of care and are assigned triage category as routine to be seen within 3 months), or who had previously been evaluated (in person or via Tele-stroke) by a stroke neurologist, were excluded. By focusing on first-time outpatient evaluations, the study targeted the population most vulnerable to delayed diagnosis and missed secondary prevention opportunities.
How were “follow-ups after admission” verified? How were Tele-stroke evaluations identified and excluded? Was symptom severity considered in triage categorization?
Follow-up after admission and telestroke evaluations were identified by chart review. A retrospective chart review was performed after the study period was completed. The following has been added tio the manuscript to reflect that process: “A structured chart review was performed of patients who attended the clinic at the end of the study period in both the surveillance and intervention phases. This formed the screening population, and, based on the inclusion and exclusion criteria described above, patients were included.”
Operational Definitions
Lines 105–118:
Define “ECG completion” more clearly (ED ECG? outpatient? done at referring site? “Missed prevention opportunity events” combine multiple unrelated constructs (diagnostic delay, delayed therapy, deterioration). This composite outcome needs justification or separation.
The following justification has been added: “The clinical outcome measures were analyzed as a composite due to the lower sample size and the lower likelihood of the event rates.”
Intervention Description
Lines 149–167: More clarity is needed on who finalized orders during the nurse call nurse alone or with neurologist authorization? Include a timeline diagram to show workflow more clearly.
During the study period, the nurse finalized the orders with the neurologist's (MK) authorization. The authorization was approved within 24 hours of the request. This has been updated in the flow diagram.
Data Collection:
Lines 128–137: Mention inter-rater consistency or whether abstraction was done independently by multiple reviewers.
Data abstraction was done independently by multiple reviewers (AH, FS, SC); we did not assess inter-rater consistency as the acquired data was objective. This has been added to line 117-118.
Statistical Analysis
Lines 192–209:
Justify use of adjusted logistic regression and specify if variables were chosen a priori. Report whether assumptions for logistic regression were met. OR “75 (95% CI 18–313)” (Line 236) appears excessively large and possibly mis-specified—likely an error needing verification.
Adjusted logistic regression was used to ascertain the actual effect of the QI study intervention cycle during the project period. The vascular imaging could have been influenced by age, sex, and triage category; hence, the odds ratio was adjusted for these physiological and system variables. We have highlighted that the variables were chosen a priori in the statistical analysis section (line 224).
The dependent variable (completion of vascular imaging) was binary, and the independent variables (QI project phase, Age, Sex and Triage category) were not multicollinear.
We have reanalyzed the median days required to complete diagnostic tests.
Initially, we performed univariate nonparametric analysis comparing medians, which showed a difference; however, when we performed univariate linear regression, only a trend toward a difference was observed. We have now revised the results section as follows: “There was a trend toward a lower median number of days required to complete the diagnostic tests in the post-intervention phase than in the pre-intervention phase in linear regression analysis (beta = -6.1, 95% CI = -12.6-0.4, p = 0.06) (adjusted OR 75, 95% CI 18-313).”
Results
Lines 212–217: Include a flow diagram showing screened, excluded, and included patients for transparency.
The numerical data on patient exclusions were not stored separately. Thus we are unable to provide the exact number and figure.
Table 1:
The p-value for referral source (ED vs PCP) is reported as 0.07, but the distribution appears meaningfully different; consider discussing potential selection/triage changes over time.
We agree with the reviewer that, although numerical differences exist, no statistical difference was observed. No referral practice pattern differences were observed during the project period. No provincial or regional changes were rolled out during the study period.
Median days to appointment shows a significant delay in the post-intervention period (p=0.008). This contradicts the manuscript’s framing of improved workflow. This requires explanation.
We noted that the median number of days to appointments increased during the intervention phase of the project, while no systemic or patient-related factors could be identified. The delay was not related to the project intervention. The nurse-led Pre-visit was organized after the patient had a clinic appointment date. We have added that to the results section (Lines 235-236) and
discussion section as well (Lines 377-381): “The median number of days to the appointment was delayed during the intervention phase of the QI project compared to the surveillance phase. While no patient-related or systemic factors could be identified, the delay was not related to the project intervention. The Nurse-led Pre-visit was organized after the patient’s clinic appointment was planned”.
Table 2 / Lines 221–238:
The OR “75” for days to investigation completion appears incorrectly calculated or misinterpreted. Provide effect-size rationale or re-analyze. The decline in Holter monitoring completion should be highlighted more prominently as a persistent system failure.
We have reanalyzed the median days required to complete diagnostic tests.
Initially, we performed univariate nonparametric analysis comparing medians, which showed a difference; however, when we performed univariate linear regression, only a trend toward a difference was observed. We have now revised the results section as follows: “There was a trend toward a lower median number of days required to complete the diagnostic tests in the post-intervention phase than in the pre-intervention phase in linear regression analysis (beta = -6.1, 95% CI = -12.6-0.4, p = 0.06) (adjusted OR 75, 95% CI 18-313).”
With respect to Holter monitoring we have added the following in the discussion section (Lines 326-328): “However, cardiac rhythm monitoring with Holter Monitoring was lower and even decreased during the intervention phase. This suggests the need for a further QI project addressing core investigations.”
Lines 239–262:
The subgroup is not randomized; clarify that findings are exploratory. Report how many patients never responded despite repeated attempts, and whether non-response introduced bias (e.g., higher stroke mimic rate among non-responders). Three urgent carotid stenosis cases were found; provide more details (degree of stenosis, symptomatic/asymptomatic).
We agree with the reviewer we have added the following to conclusion (Lines 399-400),”Since our study is a QI project and non-randomized, the results will be considered exploratory.”
We have added the following to discussion (Lines 327-329), “However, cardiac rhythm monitoring with Holter Monitoring was lower and even decreased during the intervention phase. This suggests the need for a further QI project addressing core investigations.”
We have added the following to Results section Lines 269-270: (Degree of stenosis; 1: >70% symptomatic; 2:>70% symptomatic; 3:50-70% asymptomatic)
Lines 263–266:
The statement “other contributing factors cannot be excluded” is vague. Please identify plausible confounding factors.
We have edited the statement and added the following (Lines 287-290): However, it is unclear whether the QI project intervention alone was responsible for that effect, as there are confounding variables including referral source, symptom severity and comorbidities. These variables were not systematically abstracted in the chart review.
Lines 267–280:
Include standard deviations or ranges for workload time. Clarify whether the reduction in follow-ups reached the power threshold or if the sample is underpowered.
We did not capture workload time, including time required for scheduling appointments and tests. We have added the following to lines 301-302: “We did not capture the time required for scheduling the nurse-led pre-visit, scheduling investigations and the time taken for the doppler test.”
The primary outcome measure was completion of vascular imaging in patients attending SPC. The sample size calculation was done prior to the start of QI project and the following has been added: We aimed to increase the proportion of patients completing vascular imaging prior to the clinic visit. Based on a previous regional study, approximately 61.4% receive urgent carotid imaging. To improve the rate of vascular imaging by an absolute 14%, we will need a sample size of 171 in each group, with 80% power and 5% alpha, for a total sample size of 342 patients. To accommodate no-show visits, we aimed for a sample size of 190 referrals per group.
Reference: Gladstone D et al. Urgency of carotid endartrectomy for secondary stroke prevention: results from the Registry of the Canadian Stroke Network. Stroke. 2009 Aug; 40(8): 2776-82.
Lines 281–286: Provide confidence intervals for missed prevention opportunity rates. Clarify whether any adverse events occurred during nurse-initiated therapy.
Missed opportunity rates were not our primary outcome, and hence was not captured individually. We have reported them together as a single proportion. In a future multicentre QI project, we do plan to include detailed missed opportunity analysis. No adverse events were reported during the nurse-initiated therapy. We have added this in the results section (line 289-290): “No adverse events were reported after nurse-led pre-visit clinic-initiated treatment.”
Discussion
Lines 287–336:
Discussion is well structured but occasionally over-interprets results. Avoid attributing improvements solely to the intervention when multiple system changes (e.g., new referral form) occurred concurrently.
We have tempered the discussion as suggested by the reviewer, and this has been highlighted throughout the discussion for the reviewer's consideration.
Lines 322–326:
The description of “76.8% completed all investigations” should be tempered because this subgroup inherently consisted of more complex cases.
We agree with reviewer and the discussion has been edited to reflect that.
Lines 345–347:
Provide concrete recommendations for addressing this gap, such as integration of patch monitors or community partnerships.
The following has been added to the discussion section (lines 338-343): However, cardiac rhythm monitoring with Holter Monitoring was lower and even decreased during the intervention phase. This suggests the need for a further QI project addressing core investigations. It will be important to involve community stakeholders, including increasing access to patch monitors. This would include advocacy efforts at the regional and provincial levels, given variable access to patch monitors.
Lines 338–364:
Expand discussion on temporal confounding. Discuss selection bias introduced by excluding hospital-evaluated patients. The mention of an “electronic referral form” being introduced concurrently is a major limitation and should not be minimized.
We have edited the limitation section to address the reviewers' concerns.
Conclusions
Lines 365–387:
Conclusion is sound but should reflect modest effect sizes. Avoid over-generalizing for national practice without multicenter validation.
The conclusion has been edited to temper the interpretation of the results.
- Conclusions
Krueger et al projected an increase in the number of individuals experiencing stroke by 2038 in Canada, highlighting the urgency of optimizing secondary prevention strategies. [31] A systematic review of global stroke guidelines strongly recommends establishing an etiological diagnosis and managing key risk factors. [32] This quality improvement initiative demonstrates that a structured triage algorithm combined with a well-established pre-visit process may enhance the quality, safety, and efficiency of outpatient stroke care. By shifting diagnostic review and coordination upstream in the care pathway, the intervention increased completion rates of essential investigations, such as vascular imaging, and reduced follow-up visits. It enabled more timely initiation of secondary prevention strategies. Since our study is a QI project and non-randomized, the results will be considered exploratory.
Importantly, tThis intervention was low-cost, scalable, and well-integrated into the existing clinical workflows. The model empowered the nursing staff to play a proactive role in patient care, aligning with non-financial incentives as outlined in systematic reviews [33]. This ensured that physicians had access to the key clinical data at the time of the first appointment. In doing so, the initiative reduced the time to vascular imaging and treatment initiation, an outcome of critical importance, given the narrow therapeutic window following a transient ischemic attack or minor stroke.
Although barriers remain, particularly regarding access to Holter monitoring and adoption of EMR referrals, the early impact of this intervention is promising. With 13% of pre-visit patients identified as having urgent findings requiring management before their scheduled clinic encounter, this approach is may be efficient but also potentially “stroke-preventing”.
- Future Directions
Future efforts could focus on automating the triage tools within electronic health records, integrating extended cardiac monitoring options, and exploring multicenter implementation to validate generalizability. By proactively preparing the patients before their SPC visit, the healthcare systems can reduce preventable stroke recurrence, optimize clinic capacity, and provide faster, more decisive care.
Minor Comments
- Several repeated references appear twice (e.g., references 26 and 27). Clean the bibliography.
We have checked the bibliography.
- Ensure consistent tense throughout manuscript (Methods past tense, Discussion interpretive).
This has been addressed.
- Improve figure quality; current workflow diagrams are low-resolution and difficult to read in the provided PDF.
The images have edited.
- Some sentences are excessively long and should be broken for clarity (e.g., lines 143– 148).
We have shortened the sentences.
- Correct typographical inconsistencies (missing commas, uneven spacing, mixed use of % symbols).
We have addressed these.
Reviewer 2 Report
Comments and Suggestions for Authors
Overall Assessment
This is a well-executed, single-centre quality-improvement study demonstrating that a simple, inexpensive intervention, a standardized physician triage form plus nurse-led pre-visit phone call, significantly increased pre-clinic vascular imaging completion from 61% to 75% (adjusted OR 1.77, 95% CI 1.2–2.8, p=0.01), reduced follow-up visits from 32% to 23% (p=0.052), and was highly acceptable to staff. The topic is highly relevant to everyday stroke prevention practice all over the world.
Here are some of the comments to improve the paper,
- Define exactly what “vascular imaging” includes in this clinic (CTA head/neck vs carotid duplex vs both?). The text sometimes uses “CT angiography” and sometimes “Carotid Doppler” interchangeably.
- Clarify how the triage physician decided which patients required a nurse pre-visit call (algorithm? clinical judgment?).
- Provide the actual standardized triage form as a supplementary file or appendix, or at least show the key checklist items.
- The p-value in Table 1 for “Median (IQR) Days to Appointment after Triage” is marked with an asterisk; however, no footnote explains its meaning!!
- The increase in median wait time from referral to appointment (46 → 56 days, p=0.008) is surprising and potentially concerning; therefore, explain in the discussion whether this was expected (e.g., reallocation of urgent slots) or an unintended consequence? Please add a paragraph in the discussion to clarify this more.
- Holter monitoring completion paradoxically decreased from 35.8% to 25.1% (p=0.02). This should be acknowledged and briefly discussed (access issue? lower perceived urgency?).
- Figure 1 and Figure 2 are helpful, but the red arrows and boxes are very difficult to read. Use higher contrast, add a legend, and increase the font size and figure resolution.
Author Response
Reviewer 2
- Define exactly what “vascular imaging” includes in this clinic (CTA head/neck vs carotid duplex vs both?). The text sometimes uses “CT angiography” and sometimes “Carotid Doppler” interchangeably.
Response 1: Thank you for your question. Under the operational definitions section of methods and materials, we have mentioned that vascular imaging includes either Computed Tomography angiography or Carotid Doppler. The following is mentioned in line 193-196: The primary outcome of the study was the proportion of patients who had completed vascular imaging (CT angiography for the head and neck or carotid Doppler) before their first SPC visit, as this was the investigation over which the SPC team had the most control to book a timely appointment.
- Clarify how the triage physician decided which patients required a nurse pre-visit call (algorithm? clinical judgment?)
Response 1: Thank you for your question. As you may notice in the Study setting and population section of the materials and methods, we have mentioned that the algorithm used is the Edmonton zone trans ischaemic attack and non-disabling stroke management algorithm. The triaging physician assigns a triage category based on the algorithm (risk of recurrence estimated based on the clinical presentation, completed investigations at the time of triage). Patient appointments are then booked on the basis of urgency and need for additional testing.
We have added the algorithm as supplementary file.
- Provide the actual standardized triage form as a supplementary file or appendix, or at least show the key checklist items.
Response 1: The standardized triage form is as follows. This has been added as a supplementary file
- The p-value in Table 1 for “Median (IQR) Days to Appointment after Triage” is marked with an asterisk; however, no footnote explains its meaning!!
Response 1:
We have removed the asterisk. It was to indicate significance, but the value itself is obvious.
- The increase in median wait time from referral to appointment (46 → 56 days, p=0.008) is surprising and potentially concerning; therefore, explain in the discussion whether this was expected (e.g., reallocation of urgent slots) or an unintended consequence? Please add a paragraph in the discussion to clarify this more.
Response 1:
We have added the following to the discussion:
The median number of days to the appointment was delayed during the intervention phase of the QI project compared to the surveillance phase. While no patient-related or systemic factors could be identified, the delay was not related to the project intervention. The Nurse-led Pre-visit was organized after the patient’s clinic appointment was planned.
6. Holter monitoring completion paradoxically decreased from 35.8% to 25.1% (p=0.02). This should be acknowledged and briefly discussed (access issue? lower perceived urgency?).
Response 1: We have added the following discussion:
Line 337-342:
However, cardiac rhythm monitoring with Holter Monitoring was lower and even decreased during the intervention phase. This suggests the need for a further QI project addressing core investigations. It will be important to involve community stakeholders, including increasing access to patch monitors. This would include advocacy efforts at the regional and provincial levels, given variable access to patch monitors.
Line 381-385
Limitation section
Underutilization of cardiac monitoring remains a challenge. Despite gains in vascular imaging, completion rates for Holter monitoring remained low. This reflects the logistical challenges common in the stroke clinics. Future efforts should aim to integrate more effective ambulatory cardiac monitoring options directly into the SPC workflow.
- Figure 1 and Figure 2 are helpful, but the red arrows and boxes are very difficult to read. Use higher contrast, add a legend, and increase the font size and figure resolution.
Response 1:
We have modified the figures
Figure 1: Stroke Prevention Clinic (SPC) pre-intervention workflow, triage, and patient visit. Created in BioRender. Kate, M. (2025) https://BioRender.com/buv1zgx
Figure 2: Stroke Prevention Clinic (SPC) workflow, triage, and patient visit during the intervention phase. Created in BioRender. Kate, M. (2025) https://BioRender.com/94l76nj
